# Multimodel Approaches Are Not the Best Way to Understand Multifactorial Systems

**DOI:** 10.3390/e26060506

**Published:** 2024-06-11

**Authors:** Benjamin M. Bolker

**Affiliations:** Departments of Mathematics & Statistics and Biology, McMaster University, Hamilton, ON L8S4K1, Canada; bolker@mcmaster.ca

**Keywords:** null-hypothesis significance testing, multi-model averaging, shrinkage estimators, Akaike information criterion, statistical inference

## Abstract

Information-theoretic (IT) and multi-model averaging (MMA) statistical approaches are widely used but suboptimal tools for pursuing a multifactorial approach (also known as the method of multiple working hypotheses) in ecology. (1) Conceptually, IT encourages ecologists to perform tests on sets of artificially simplified models. (2) MMA improves on IT model selection by implementing a simple form of shrinkage estimation (a way to make accurate predictions from a model with many parameters relative to the amount of data, by “shrinking” parameter estimates toward zero). However, other shrinkage estimators such as penalized regression or Bayesian hierarchical models with regularizing priors are more computationally efficient and better supported theoretically. (3) In general, the procedures for extracting confidence intervals from MMA are overconfident, providing overly narrow intervals. If researchers want to use limited data sets to accurately estimate the strength of multiple competing ecological processes along with reliable confidence intervals, the current best approach is to use full (maximal) statistical models (possibly with Bayesian priors) after making principled, a priori decisions about model complexity.

Modern scientific research often aims to quantify the effects of multiple simultaneously operating processes in natural or human systems. Some examples from my own work in ecology and evolution consider the effects of herbivory and fertilization on standing biomass [1]; the effects of bark, wood density, and fire on tree mortality [2]; or the effects of taxonomic and genomic position on evolutionary rates [3]. This multifactorial approach [4] complements, rather than replacing, the traditional hypothesis-testing or strong-inferential framework [5,6,7]. (While there is much interesting debate over the best methods for gathering evidence to distinguish among two or more particular, intrinsically discrete hypotheses [8], that is not the focus of this paper.) Such attempts to quantify the magnitude or importance of different processes also differ from predictive modeling, which dominates the fields of machine learning and artificial intelligence [9]. The prediction and quantification of process strength are closely related—if we can accurately predict outcomes over a range of conditions, then we can also predict the effects of changes in those conditions, and hence infer the strengths of processes, if the changes we are trying to predict are adequately reflected in our training data. However, predictive modelers are usually primarily concerned with predictions within the natural range of conditions, which may not provide us enough information to reliably make inferences about processes. The paper focuses on statistical modeling for estimation and inference, rather than prediction.

A standard approach to analyzing multifactorial systems, particularly common in ecology, is as follows: (1) Construct a full model that encompasses as many of the processes (and their interactions) as is feasible. (2) Fit the full model and make sure that it describes the data reasonably well (e.g., by examining model diagnostics and by ensuring that the level of unexplained variation is not unacceptably large). (3) Construct possible submodels of the full model by setting the subsets of parameters to zero. (4) Compute the information-theoretic measures of quality, such as the Akaike or Bayesian/Schwarz information criteria, for every submodel. (5) Use multi-model averaging (MMA) to estimate model-averaged parameters and confidence intervals (CIs), and possibly draw conclusions about the importance of different processes by summing the information-theoretic weights [10]. I argue that this approach, even if used sensibly as advised by proponents of the approach (e.g., with reasonable numbers of candidate submodels), is a poor way to approach estimation and inference for multifactorial problems.

For example, suppose we want to understand the effects of ecosystem-level net primary productivity and fire intensity on species diversity (a simplified version of the analysis done in [11]). The model-comparison or model-averaging approach would construct five models: a null model with no effects of either productivity or fire, two single-factor models, an additive model, and a full model allowing for interactions between productivity and fire. We would then fit all of these models and model-average their parameters, and derive model-averaged confidence intervals.

The goal of a multifactorial analysis is to tease apart the contributions of many processes, all of which we believe are affecting our study system to some degree. If our scientific questions are (something like) “How important is this factor, in an absolute sense or relative to other factors?” (or equivalently, “How much does a change in this factor change the system in absolute or relative terms?”), rather than “Which of these factors are having any effect at all on my system?”, why are we working so hard to fit many models of which only one (the full model) incorporates all of the factors? If we do not have particular, a priori discrete hypotheses about our system (such as “process *A* influences the outcome but process *B* has no effect at all”), why does so much of our data-analytic effort go into various ways to test between, or combine and reconcile, multiple discrete models? In software development, this is called an “XY problem” (http://www.perlmonks.org/?node=XY+Problem, accessed on 30 May 2024): rather than thinking about the best way to solve our real problem *X* (understanding multifactorial systems), we have become bogged down in the details of how to make a particular tool, *Y* (multimodel approaches), provide the answers we need. Most critiques of MMA address technical concerns such as the influence of unobserved heterogeneity [12] or criticize the misuse of information-theoretic methods by researchers [13,14], but do not ask why we are comparing discrete models in the first place.

In contrast with averaging across discrete hypotheses or treating a choice of discrete hypotheses as an end goal, fitting and comparing multiple models as a step in a null-hypothesis significance testing (NHST) procedure is defensible. In the biodiversity analysis described above, we might fit the full model and then assess the significance of individual terms by comparing the fit of the full model to models with those terms dropped (taking particular care with the interpretation of dropping a lower-level effect in models with interactions, e.g., see [15]). While much maligned, NHSTs are a useful part of data analysis—not to decide whether we really think a null hypothesis is false (they almost always are), but to see if we can distinguish signal from noise. Another interpretation is that NHSTs can test whether we can reliably determine the direction of effects—that is, not whether the effect of a predictor on some process is zero, but whether we can tell unequivocally that it has a particular sign—positive or negative [16,17].

However, researchers using multimodel approaches are not fitting one-step-reduced models to test hypotheses; rather, they are fitting a wide range of submodels, typically in the hope that model choice or multimodel averaging will help them deal with insufficient data in a multifactorial world. If we had enough information (even Big Data does not always provide the information we need [18], we could fit only the full model, drawing our conclusions from the estimates and CIs with all of the factors considered simultaneously. But we nearly always have too many predictors, and not enough data; we do not want to overfit, (which will inflate our CIs and *p*-values to the point where we cannot tell anything for sure), but at the same time we are afraid of neglecting potentially important effects.

Stepwise regression, the original strategy for separating signals from noise, is now widely deprecated because it interferes with correct statistical inference [19,20,21,22]. Information-theoretic tools mitigate the instability of stepwise approaches, allow for the simultaneous comparison of many non-nested models, and avoid the stigma of NHST. A further step forward, multi-model averaging [10], accounts for model uncertainty and avoids focusing on a single best model. Some forms of model averaging provide shrinkage estimators; averaging the strength of effects between models where they are included and models where they are absent adjusts the estimated effects toward zero [14]. More recently, model averaging is experiencing a backlash, as studies point out that multimodel averaging may run into trouble when variables are collinear and/or have differential levels of measurement error [23], when we are careless about the meaning of main effects in the presence of interactions, when we average model parameters rather than model predictions [14], or when we use summed model weights to assess the relative importance of predictors ([14,24]; but cf. [25]).

Freckleton [23] makes the point that model averaging will tend to shrink the estimates of multicollinear predictors toward each other, so that estimates of weak effects will be biased upward and estimates of strong effects will be biased downward. This is an unsurprising (in hindsight) consequence of shrinkage estimation. With other analytical methods such as lasso regression, or selection of a single best model by AIC, the weaker of two correlated predictors, or more precisely the one that appears weaker based on the available data, could be eliminated entirely, leading all of its effects to be attributed to the stronger predictor. Researchers often make a case for dropping correlated terms in this way because collinearity of predictors inflates parameter uncertainty and complicates interpretation. However, others have repeatedly pointed out that collinearity is a problem of intrinsic uncertainty—we are simply missing the data that would tell us which combination of collinear factors really drives the system. The confidence intervals of parameters from a full model estimated by regression or maximum likelihood will correctly identify this uncertainty; modeling procedures that automatically drop collinear predictors (by model selection or sparsity-inducing penalization) not only fail to resolve the issue, but can lead to inaccurate predictions based on new data [26,27,28,29]. A full model might (correctly) tell us we cannot confidently assess whether either productivity or fire decrease or increase species diversity, because their estimated effects are strongly correlated. However, by comparing the fit of the full model to one that dropped both productivity and fire, we could conclude that their joint effect is highly significant.

In ecology, information criteria were introduced by applied ecologists who were primarily interested in making the best possible predictions to inform conservation and management; they were less concerned with inference or quantifying the strength of underlying processes [10,30,31]. Rather than using information criteria as tools to identify the best predictive model, or to obtain the best overall (model-averaged) predictions, most current users of information-theoretic methods use them either to quantify variable importance, or, by multimodel averaging, to have their cake and eat it too—to avoid either over- or underfitting while quantifying the effects in multifactorial systems. There are two problems with this approach—one conceptual and one practical.

The conceptual problem with model averaging reflects the original sin of unnecessarily discretizing a continuous model space. When we fit many different models as part of our analytical process (based on selection or averaging), the models are only a means to an end; despite the claims of some information-theoretic modelers, we are not really using the submodels in support of the method of multiple working hypotheses as described by Chamberlin [32]. For example, Chamberlin argued that in teaching about the origin of the Great Lakes, we should urge students “to conceive of three or more great agencies [pre-glacial erosion, glacial erosion, crust deformation] working successively or simultaneously, and to estimate how much was accomplished by each of these agencies”. Chamberlin was not suggesting that we test which individual mechanism or combination of mechanisms fits the data best (in whatever sense), but instead that we acknowledge that the world is multifactorial. In a similar vein, Gelman and Shalizi [33] advocate “continuous model expansion”, creating models that include all components of interest (with appropriate Bayesian priors to constrain the overall complexity of the model) rather than selecting or averaging across discrete sets of models that incorporate subsets of the processes.

Here, I am not concerned whether ‘truth’ is included in our model set (it is not), and how this matters to our inference [34,35]. I am claiming the opposite, that our full model—while certainly not the true model—is usually the closest thing we have to a true model. This claim seems to contradict the information-theoretic result that the best approximating model (i.e., the minimum-AIC model) is expected to be closest to the true (generating) model in a predictive sense (i.e., it has the smallest expected Kullback–Leibler distance) [36]. However, the fact that excluding some processes allows the fitted model to better match the observation does not mean that we should believe these processes are not affecting on our system—just that, with the available data, dropping terms will provide us better predictions than keeping the full model. If we are primarily interested in prediction, or in comparing qualitatively different, possibly non-nested hypotheses [37], information-theoretic methods match our goals well.

The technical problem with model averaging is its computational inefficiency. Individual models can take minutes or hours to fit, and we may have to fit dozens or scores of sub-models in the multi-model averaging process. There are efficient tools available for fitting “right-sized” models that avoid many of the technical problems of model averaging. Penalized methods such as ridge and lasso regression [38] are well known in some scientific fields; in a Bayesian setting, informative priors centered at zero have the same effect of regularizing—pushing weak effects toward zero and controlling model complexity (more or less synonymous with the shrinkage of estimates described above) [39]. Developed for optimal (predictive) fitting in models with many parameters, penalized models have well-understood statistical properties; they avoid the pitfalls of model-averaging correlated or nonlinear parameters; and, by avoiding the need to fit many sub-models in the model-averaging processes, they are much faster. (However, they may require a computationally expensive cross-validation step in order to choose the degree of penalization.) Furthermore, penalized approaches underlie modern nonparametric methods such as additive models and Gaussian processes that allow models to expand indefinitely to match the available data [40,41].

Penalized models have their own challenges. A big advantage of information-theoretic methods is that, like wrapper methods for feature selection in machine learning [42], we can use model averaging, as long as we can fit component models and extract the log-likelihood and number of parameters—we never need to build new software. Although powerful computational tools exist for fitting penalized versions of linear and generalized linear models (e.g., the glmnet package for R) and mixed models (glmmLasso), quantile regression [43], software for some more exotic models (e.g., zero-inflated models, models for censored data) may not be readily available. Fitting these models requires the user to choose the strength of penalization. This process is conveniently automated in tools like glmnet, but correctly assessing the out-of-sample accuracy (and hence the correct level of penalization) is tricky for data that are correlated in space or time [44,45]. Penalization (or regularization) can also be achieved by imposing Bayesian priors on subsets of parameters [46], but this converts the choice of strength of penalization to a similarly challenging choice of appropriate priors.

Finally, frequentist inference (computing *p*-values and CIs) for parameters in penalized models—one of the basic outputs we want from a statistical analysis of a multifactorial system—is a current research problem; statisticians have proposed a variety of methods [47,48,49,50], but they typically make extremely strong asymptotic assumptions and are far from being standard options in software. Scientists should encourage their friends in statistics and computer science to build tools that make penalized approaches easier to use.

Statisticians derived confidence intervals for ridge regression long ago [51]—surprisingly, they are identical to the confidence intervals one would have achieved from the full model without penalization. Wang and Zhou [52] similarly proved that model-averaging CIs derived as suggested by Hjort and Claeskens [53] are asymptotically (i.e., for arbitrarily large data sets) equivalent to the CIs from the full model. Analytical and simulation studies [54,55,56,57,58,59] have shown that a variety of alternative methods for constructing CIs are overoptimistic, i.e., that they generate too-narrow confidence intervals with coverage lower than the nominal level. Simulations from several of the studies above show that MMA confidence intervals constructed according to the best known procedures typically include the true parameter values only about 80% or 90% of the time. In particular, Kabaila et al. [58] say that constructing CIs that take advantage of shrinkage but still achieve correct coverage will be very difficult to achieve using model averaged confidence intervals. (The only examples I have been able to find of MMA confidence intervals with close to nominal coverage are from Chapter 5 of [10].) In short, it seems difficult to find model-averaged confidence intervals that compete successfully with the standard confidence interval based on the full model.

Free lunches do not exist in statistics, any more than anywhere else. We can use penalized approaches to improve prediction accuracy without having to sacrifice any input variables (by trading bias for variance), but the only known way to gain statistical power for testing hypotheses, or narrowing our uncertainty about our predictions, is to limit the scope of our models a priori [19], to add information from pre-specified Bayesian priors (or equivalent regularization procedures), or to collect more data. Burnham and Anderson [60] defined a “savvy” prior that reproduces the results of AIC-based multimodel averaging in a Bayesian framework, but it is a weak conceptual foundation for understanding multifactorial systems. Because it is a prior on discrete models, rather than on the magnitude of continuous parameters that describe the strength of different processes, it induces a spike-and-slab type prior on parameters that assigns a positive probability to the unrealistic case of a parameter being exactly zero; furthermore, the prior will depend on the particular set of models being considered.

Multimodel averaging is probably most popular in ecology (in May 2024, Google Scholar returned ≈ 65,000 hits for “multimodel averaging” alone and 31,000 for “multimodel averaging ecology”). However, multifactorial systems—and the problems of approaching inference through comparing and combining discrete models that consider artificially limited subsets of the processes we know are operating — occur throughout the sciences of complexity, those involving biological and human processes. In psychology, economics, sociology, epidemiology, ecology, and evolution, every process that we can imagine has some influence on the outcomes that we observe. Pretending that some of these processes are completely absent can be a useful means to an inferential or computational end, but it is rarely what we actually believe about the system (although see [13] for a counterargument). We should not let this useful pretense become our primary statistical focus.

If we have sensible scientific questions and good experimental designs, muddling through with existing techniques will often provide reasonable results [61]. But researchers should at least be aware that the roundabout statistical methods they currently use to understand multifactorial systems were designed for prediction, or the comparison of discrete hypotheses, rather than for quantifying the relative strength of simultaneously operating processes. When prediction is the primary goal, penalized methods can work better (faster and with better-understood statistical properties) than multimodel averaging. When estimating the magnitude of effects or judging variable importance, penalized or Bayesian methods may be appropriate—or we may have to go back to the difficult choice of focusing on a restricted number of variables for which we have enough to data to fit and interpreting the full model.

## Data Availability

There is no data in the article.

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
