# Peer review of "Multimodel Approaches Are Not the Best Way to Understand Multifactorial Systems"

_entropy, 2024, doi:10.3390/e26060506_

Round 1

Reviewer 1 Report

Comments and Suggestions for Authors

Reviewer 2 Report

Comments and Suggestions for Authors

There’s a lot to like about this paper. Who doesn’t enjoy a polemic? Who doesn’t want a simple answer to a complicated problem. Ultimately, however, I don’t think this paper works as a standalone paper. Either, it could serve as the basis of a read paper with rejoinders and further discussion. Or it could be expanded under the more honest title: “There is no best way to understand multifactorial systems”.

Doing statistics is hard. Learning from data is hard. To deny this is to show that one has never had to analyse real observational data and that one thinks generations of statisticians have been severely misguided as to what they should be doing.

The view that the best approach is “to use full (maximal) statistical models” is, as the author is no doubt fully aware, completely misleading. It fails spectacularly in two very common scenarios: (1) when there are correlated explanatory variables, and (2) when the effect on the outcome of interest is nonlinear. To take the author’s own example: precipitation and temperature are correlated variables whose effect is often nonlinear. When they are correlated, the full model can easily be too complicated and inferred effects can have the wrong sign and/or inflated CIs; when their effect is nonlinear, the full model is too simple.

And this raises another issue with the paper: the class of models the author seems to be contemplating is rather limited. Why not use generalised additive models? Random forests? There is a whole raft of approaches to prevent overfitting, to determine the relative importance of explanatory variables, etc that has real relevance to the question of how to understand multifactorial systems.

Finally, there is no discussion (or criticism) in the paper of the Gelman & Hill approach to model building. I think this has particular relevance to the ecological applications that the author has in mind.

Minor comments:

* An actual working example of the author’s claims would have aided this paper enormously. 

* Line 34: Please can we stop advocating that computing R^2 is a measure of whether our model “describes the data reasonably well”. Perhaps more important here is to stress the importance of the checking of *model assumptions*.

* Line 103: I am not sure what is meant by “unnecessarily discretizing a continuous world” in this context. Is the author suggesting model space is continuous? But isn’t that what MMA is trying to recover? Or is the author saying that it is a category error to imagine the components of a multifactorial system have independent effects? But in that case a multifactorial model can only ever be a reflection of the imaginative power of the researcher. Who is to say what the “full statistical model” is?

Miscellaneous:

* Line 193: The author needs to state when this Google Scholar search was performed.

Reviewer 3 Report

Comments and Suggestions for Authors

Round 2

Reviewer 1 Report

Comments and Suggestions for Authors

Revised ms seems fine.  The paragraph on multicollinearity is a good addition.

Reviewer 3 Report

Comments and Suggestions for Authors

The author has addressed my questions and made changes that I believe improved the paper to the point of making it publishable.